# YOLOv5s-CA: A Modified YOLOv5s Network with Coordinate Attention for Underwater Target Detection

**DOI:** 10.3390/s23073367

**Published:** 2023-03-23

**Authors:** Ge Wen, Shaobao Li, Fucai Liu, Xiaoyuan Luo, Meng-Joo Er, Mufti Mahmud, Tao Wu

**Affiliations:** 1School of Electrical Engineering, Yanshan University, Qinhuangdao 066004, China; 2Institute of Artificial Intelligence and Marine Robotics, College of Marine Electrical Engineering, Dalian Maritime University, Dalian 116026, China; 3Department of Computer Science, Computing and Informatics Research Centre, Medical Technologies Innovation Facility of Nottingham Trent University, Nottingham NG11 8NS, UK; 4Department of Frontier & Innovation Research, Wuhan Second Ship Design & Research Institute, Wuhan 430205, China

**Keywords:** underwater target detection, deep learning, YOLO neural network, Coordinate Attention

## Abstract

Underwater target detection techniques have been extensively applied to underwater vehicles for marine surveillance, aquaculture, and rescue applications. However, due to complex underwater environments and insufficient training samples, the existing underwater target recognition algorithm accuracy is still unsatisfactory. A long-term effort is essential to improving underwater target detection accuracy. To achieve this goal, in this work, we propose a modified YOLOv5s network, called YOLOv5s-CA network, by embedding a Coordinate Attention (CA) module and a Squeeze-and-Excitation (SE) module, aiming to concentrate more computing power on the target to improve detection accuracy. Based on the existing YOLOv5s network, the number of bottlenecks in the first C3 module was increased from one to three to improve the performance of shallow feature extraction. The CA module was embedded into the C3 modules to improve the attention power focused on the target. The SE layer was added to the output of the C3 modules to strengthen model attention. Experiments on the data of the 2019 China Underwater Robot Competition were conducted, and the results demonstrate that the mean Average Precision (mAP) of the modified YOLOv5s network was increased by 2.4%.

## 1. Introduction

The ocean contains abundant resources, such as oil, natural gas, seafood, etc., which are being explored to satisfy the fast development of human society. However, the harsh marine environment is not friendly to human beings, and marine vehicles equipped with underwater target detection techniques are increasingly utilised for various marine activities, such as mine exploration, underwater aquaculture, marine rescue, etc. [1,2].

Most of the existing object detection methods are mainly applied to terrestrial environments. Benefiting from large quantities of high-quality training samples, it is relatively easy to obtain a high-precision network model for target detection. However, model training for underwater target detection is much more difficult due to the complex underwater environments (e.g., turbid water, uneven lighting, ocean current interference, etc. [3]) and insufficient datasets.

The existing object detection methods can be generally divided into two categories: One is target detection based on hand-crafted features, and another is target detection based on deep learning [4]. The former mainly relies on the manual feature design of the targets to be detected. These features are often intuitive, such as colour, texture, shape, etc., and people can directly identify these features. Some common detection methods based on hand-crafted features include Local Binary Pattern (LBP) [5], Scale-Invariant Feature Transform (SIFT) [6], Histogram of Oriented Gradients (HOG) [7], etc. Villon et al. combined HOG and SVM to detect coral reef fish in the collected images to solve the accuracy degradation caused by target occlusion and overlap in underwater images [8]. Although the target detection algorithm based on hand-crafted features has high accuracy for the specific targets, it largely depends on the designer’s knowledge of related fields. Moreover, the subjective views of designers also affect the design of features to a certain extent. In the feature design process, there is a need to consume a lot of manpower and material resources, which inevitably causes certain losses.

In recent years, deep learning research has gradually gained social recognition. The application of deep learning in the field of target detection has gradually become a research hotspot. Compared with target detection based on hand-crafted features, target detection based on deep learning has the advantages of wider application fields, convenient design, and simple dataset production, which can save a lot of manpower and material resources. Target detection approaches based on deep learning can be divided into region proposal-based and regression-based algorithms. In region proposal-based target detection algorithms, a Region Proposal Network (RPN) is included in the model to generate the candidate object bounding boxes to improve object detection and classification. Some typical region proposal-based target detection models include Fast R-CNN [9], Faster R-CNN, Mask R-CNN [10], R-FCN [11], etc. Zeng et al. combined the Faster R-CNN network and the adversarial occlusion network to construct a new network named Faster R-CNN-AON network [12]. The network works well in suppressing the overfitting that can occur during model training. Song et al. improved the Mask R-CNN network, and the improved model could perform well in target detection in complex underwater environments [13]. Although the proposed target detection algorithm based on region proposal has relatively high detection accuracy, it is quite time-consuming for underwater target detection and is not suitable for real-time detection tasks.

Compared with the target detection approaches based on region proposal, the ones based on regression do not have a Region Proposal Network (RPN) but directly generate the corresponding predicted boxes on the input image to detect the target. Therefore, regression-based target detection approaches have faster detection speed and lower hardware requirements and are suitable for portable devices. Some typical regression-based target detection models are YOLO [14], YOLO9000 [15], YOLOv3 [16], YOLOv4 [17], SSD [18], etc. Chen et al. proposed an improved YOLOv4 model named YOLOv4-UW [19]. The authors removed the large-sized output layer and the SPP structure from the original model and used the deconvolution module to replace the original upsampling module. The improved model increases the speed of the detection of specific objects. Yao et al. improved the SSD model [20]. The authors designed two residual units to replace the depth separable convolution in the original model as the feature extractor of the model, which greatly reduced the number of parameters of the model, thus speeding up the training speed of the model. Therefore, regression-based detection algorithms are less consuming and are suitable for real-time detection tasks. However, their detection accuracy is relatively low. Moreover, due to the limitations of the hardware equipment of underwater robots, most of the existing underwater target detection tasks are still dominated by regression-based target detection algorithms.

Algorithm detection accuracy has been improved without sacrificing detection speed with improvements in the original regression-based target detection algorithms. Among the various improvements, embedding the attention module into the model is one of the most effective methods. The principle of the attention mechanism is to focus most of the computer’s computing power on important objects and ignore the unimportant parts of the image, such that the detection model can more efficiently detect the objects in the picture. Li et al. aimed to detect small and blurry targets in infrared images [21]. Most of the targets are small and blurry. They embedded the SK attention module in the original YOLOv5 model to obtain good accuracy in the detection of blurred objects in complex environments. The YOLOv5 algorithm is a regression-based target detection model with multiple modes. Due to its simple model and few parameters, when applied to underwater target detection, it can achieve good real-time performance and can be easily deployed in various portable underwater robots. However, there are still no instances of applying the YOLOv5 model to underwater target detection. In this work, we embedded the attention mechanism into the YOLOv5 model to improve the detection accuracy in complex underwater environments without greatly reducing the detection speed.

The main contributions of this work are summarised as follows: Firstly, the number of bottlenecks in the first C3 module of the model was increased from the original one to three to improve the capability of shallow feature extraction; thereby, the model can collect more subtle features. Next, some C3 modules were modified, and the CA attention module was embedded in the C3 modules to improve the model attention focused on the region of interest; thereby, the model can concentrate the computing power on the region of interest. Finally, the SE layer was added to the specific position of the model to further strengthen the model attention focused on the region of interest.

The rest of this paper is as follows: In Section 2, the original YOLOv5s model is briefly introduced, and the corresponding innovations proposed in this paper are described. Section 3 describes the experimental platform, parameter settings, etc., and the experimental results are recorded and analysed. Finally, Section 4 concludes this paper with some prospective related works.

## 2. Modified YOLOv5s Model

In this section, an overview of the YOLOv5s network is first presented. Next, based on the basic YOLOv5s structure, the CA and SE modules employed to modify the YOLOv5s model, named YOLOv5s-CA, to improve the accuracy of underwater target detection are introduced. Figure 1 shows the diagram of the process for applying the modified YOLOv5s model to underwater target detection.

### 2.1. Basic YOLOv5s Model

The YOLOv5 model is a regression-based target detection model released by Glenn Jocher in 2020. The YOLOv5 model was developed based on target detection models such as YOLOv3 and YOLOv4. Compared with the previous models, the YOLOv5 model improves the detection accuracy of the model while maintaining the detection speed. The YOLOv5 model has four structures: YOLOv5s, YOLOv5m, YOLOv5l, and YOLOv5x. The difference among these four structures is mainly the difference in the number of convolution kernels and bottlenecks in specific parts. The parameters of the four models increase in turn, and accordingly, the detection accuracy of the model also increases in turn, but the detection speed of the model decreases accordingly. In the actual underwater environment, the hardware equipment of underwater robots is limited. Thus, strict restrictions on the model size are necessary, which motivated this work to use the YOLOv5s model as the experimental object.

The YOLOv5s model is mainly divided into three parts: backbone network, neck network, and detect network. The backbone network is a convolutional neural network to generate feature maps of different sizes. The backbone network comprises the Focus module, the Conv module, the C3 module, and the Spatial Pyramid Pooling (SPP) module. The neck network adopts the structure of Feature Pyramid Network (FPN) + Path Aggregation Network (PAN). It fuses low-level spatial features with high-level semantic features generated by the backbone network using bidirectional fusion. It then inputs the generated feature maps into the detect network. The detect network uses anchor boxes to operate on the input feature map to generate detection boxes to indicate the target type, location, and confidence in the image. Figure 2 shows the overall network structure of the YOLOv5s model.

### 2.2. Attention Mechanism

In humans, when observing objects, the human visual system tends to focus on the most important objects while ignoring the parts of the line of sight that are not important for identifying objects [22]. The attention mechanism [23] is similar to that of the human visual system, as it can tell the object detection model which objects are important and where they are located in the acquired image [24]. In existing studies, a large number of researchers have embedded the attention mechanisms into deep neural networks and achieved good experimental results [25]. Some common examples of embedding the attention mechanisms into the deep neural networks include object classification [26], image segmentation [27], target detection [28], etc.

Some typical attention modules include the SK module [29], the CBAM module [30], the TA module [31], etc. The CBAM module, proposed in 2018, is an attention module that can be incorporated into feed-forward convolutional neural networks. It consists of two independent sub-modules, the Channel Attention Module (CAM) and the Spatial Attention Module (SAM). The CAM focuses on the distribution relationship of feature map channels, while the SAM allows the neural network to pay attention to the regions of the image that are the most relevant for classification. By considering both spatial and channel attention, the CBAM module can enhance the performance of the model.

The SK module, proposed in 2019, is another Channel Attention Module commonly used in vision models. It utilizes a convolution kernel mechanism that assigns different levels of importance to convolution kernels for different input images. By using convolution kernels of different scales, the SK module can extract features from input feature maps and generate channel attention information to enhance the channel features without changing the dimensionality.

The Squeeze-and-Excitation (SE) network module, proposed by Momenta in 2017, is another attention module that won the image recognition championship in the ImageNet competition [32]. The SE module models the interdependence among feature channels and learns the importance of each channel to weight the features, highlighting important features and suppressing unimportant ones. The SE module improves the expression ability of the model, enhancing the detection ability of blurred images while remaining lightweight and not imposing a significant computational burden on the model. It can be easily integrated into various network frameworks and achieves significant performance improvements at a slight computational cost. Embedding the SE module in the appropriate position of a target detection model can greatly improve the detection accuracy of the model. Figure 3 shows the structure of the SE attention module.

The Coordinate Attention (CA) attention module [33] is an attention module proposed by Hou et al. in 2021. Compared with the previous modules, such as the CBAM module, the CA attention module has made satisfactory progress in deep neural networks. The CA attention module embeds the target location information in the image into the channel attention so that the detection model can better locate the target and detect it. The specific operation of the CA attention module can be divided into coordinate information embedding and Coordinate Attention generation. Instead of compressing the feature tensor into a single feature vector with 2D global pooling, the CA attention module decomposes the input feature tensor into two 1D feature coding processes in two different directions (horizontal and vertical directions), respectively. These two 1D feature encoding processes allow the model to capture long-range dependencies among channels along one direction and maintain the location information of objects along the other. At this point, the target location information is saved to the generated attention map and then added to the input feature map with a multiplication operation for the next convolution operation. Figure 4 shows the structure of the CA attention module.

### 2.3. YOLOv5s-CA Model

In the complex underwater environment, since the collected images are often blurred, if the target detection model is directly applied to target detection tasks, the detection accuracy is greatly reduced. By expanding the number of Bottleneck modules in the first C3 module, the capability of low-level feature extraction can be improved, thereby improving the detection performance of blurred objects. Moreover, the attention module is embedded into the model, which can further enhance the performance of the model in the detection of blurred or small targets.

#### 2.3.1. Extended Bottleneck Module

There are many convolutional layers in the backbone network of YOLOv5s for the feature extraction of targets. As the network performs multiple convolutions on an image, the high-level features of the image are extracted for subsequent processing. These high-level features contain plenty of semantic information but have very low resolution, resulting in low accuracy in the detection of blurred objects. In contrast, in the shallow modules of the network, the model is able to perform better feature extraction of low-level features and obtain better resolution. For small and weak targets with few features in underwater images, deep convolution may cause difficulty in feature extraction or even feature loss. To maximise the features of blurred objects in underwater images, it is necessary to fully use the high-resolution features extracted by the shallow network. Therefore, it is reasonable to process the shallow modules of the YOLOv5s model. We expanded the number of bottlenecks in the first C3 module from the original one to three such that the number of bottlenecks in the first C3 module is the same as the number of bottlenecks in the following C3 modules. By expanding the number of bottlenecks in the first C3 module, the performance in extracting low-level features can be improved. Figure 5 shows the schematic diagram of the first C3 module here improved.

As can be seen in Figure 5, in the first C3 module, there are two branches that perform feature extraction. Then, the obtained features merge to obtain a richer feature combination. By increasing the number of bottlenecks in the first C3 module, the extraction of low-level features can be improved without significantly increasing the complexity of the model; as a result, the model can better detect blurred targets.

#### 2.3.2. Improved C3 Module

By integrating the attention mechanism into the target detection model, the model ability to focus on objects in the image can be enhanced, enabling it to allocate most of the computational resources to the target location. In this paper, a novel attention module, namely, the CA attention module, is proposed. This module was designed to preserve the positional information of the target object in the image and incorporate it into the model convolution operation.

This paper combined the CA attention module and the C3 module of the YOLOv5s model to replace the original C3 module, aiming to improve the performance of the detection model in the detection of blurred underwater targets. The modification combination is to add a CA module between the Conv module and the Bottleneck module in the trunk branch of the C3-True module to improve the capability of performing shallow feature extraction. The second is to use the CA module to replace the Bottleneck module in the trunk branch of the C3-False module. The second combination method is mainly adopted to reduce the number of parameters of the model while improving the detection accuracy of the model without greatly decreasing the detection speed.

The appropriate reduction ratio in the original CA attention module is used to reduce the number of channels. In this paper, we modified the value of the reduction ratio from the original 32 to 8 to reduce the number of parameters to a certain extent. The schematic diagram of the first combination mode, named the C3-1 module, is shown in Figure 6. The schematic diagram of the second combination mode, called the C3-2 module, is shown in Figure 7.

#### 2.3.3. Improved Backbone Network

Due to the complexity of the underwater environment, the quality of the collected underwater pictures is often not high, and so is that of the targets in the pictures. In addition, a variety of targets need to be detected in one picture. Therefore, the detection model should extract the features of each target from different complex feature maps. Thus, we embedded the SE (Squeeze-and-Excitation) attention module into a suitable position in the backbone network of the YOLOv5s model to adaptively adjust the channel weights according to the convolutional input without greatly increasing the model complexity.

In this paper, the SE module was added to the output of some C3 modules in the backbone network to enhance the feature extraction capability of the YOLOv5s model. The experiments conducted in this study revealed that incorporating an SE attention module following the C3 module can enhance the model ability to concentrate on the target, as well as extract more comprehensive features from the feature map obtained from the C3 module. As a result, the model can extract richer features. This combination allows the model to recalibrate the channels on any convolutional layer before the features are passed to the subsequent convolutional layers. With this improvement, the detection model can suppress the unimportant objects in the image well and enhance the performance in the detection of blurred objects.

To efficiently extract the shallow features of images, this paper expanded the number of bottlenecks in the first C3 module. Moreover, by embedding the CA attention module into some C3 modules, the C3 modules were improved, such that the model can improve the attention focused on the important objects in the image and reduce the computational cost. Finally, the SE attention module was embedded in the appropriate position of the backbone network, such that the model can improve the attention focused on important objects in the image while ignoring the unimportant positions in the image. Figure 8 shows the structural diagram of the modified backbone network. We named the modified YOLOv5s network the YOLOv5s-CA model. Corresponding verification experiments were carried out, as reported in the next section, to verify the effectiveness of the method proposed in this paper.

## 3. Experimental Setup and Results

### 3.1. Dataset and Training Platform

The dataset in this paper comes from the underwater target detection group in the 2019 China Underwater Robot Competition. This dataset contains a total of 3701 underwater pictures. Since the dataset uses the VOC data format, it was necessary to label the original pictures. In this paper, the LabelImg program was used to label the original pictures, and their format was modified into the YOLO format before it could be applied to the training of the YOLOv5s-CA model. In the prepared dataset, 2900 images were randomly selected as the training set; a total of 400 images were randomly selected as the validation set; and 100 images were randomly selected as the test set. The dataset in this paper contains four categories: echinus, starfish, scallop, and holothurian.

This experiment was implemented with the operating system of CentOS Linux release 7.9.2009 (Core), Python3.8, CUDA11.1, and pytorch1.8.0. Hardware equipment included AMD EPYC 7402 48@2.8 HZ CPU, NVIDIA RTX 3090 GPU, 24 GB memory, and 60 GB RAM.

### 3.2. Evaluation Index

In this paper, two indicators for detection accuracy and speed were used to evaluate the model performance. Precision (*P*) and recall (*R*) are defined as follows: (1)$$\begin{array}{cc} \; & P=\frac{TP}{FP+TP} \end{array}$$(2)$$\begin{array}{cc} \; & R=\frac{TP}{FN+TP} \end{array}$$
where True Positive (TP) means the number of correctly detected underwater targets, False Positive (FP) indicates the number of backgrounds falsely detected as objects, and False Negative (FN) means the number of objects falsely detected as backgrounds. Precision and recall affect each other; thus, they cannot be directly used to evaluate model performance.

In the field of target detection, the mAP (mean Average Precision) is usually used as the evaluation index for detection accuracy, as it refers to the average accuracy of multiple categories. The AP (Average Precision) value is the detection accuracy of the model in a single category and is a comprehensive evaluation index based on precision and recall. By varying the confidence threshold, different levels of precision and recall can be obtained. By testing the model with various confidence thresholds ranging from 0 to 1, a set of precision and recall values can be obtained. Plotting these corresponding values on a coordinate axis results in a precision–recall (P-R) curve. The area under the curve (AUC) of the P-R curve corresponds to the Average Precision (AP) value of the class being evaluated. The mAP is the average AP value of all categories. The mAP value is between 0 and 1. A high mAP value indicates that the model has high detection accuracy in multiple species. AP and mAP are defined as follows: (3)$$\begin{array}{cc} \; & AP=\int_{0}^{1}P(R)\;dR \end{array}$$(4)$$\begin{array}{cc} \; & mAP=\frac{1}{C}\sum_{i=1}^{C}AP\left(i\right) \end{array}$$
where *C* means the number of underwater target species.

FPS (Frame Per Second) is usually used as the evaluation index for detection speed, as it refers to the number of frames processed by the model per second. The higher the FPS value is, the better the real-time performance of the model is. In this experiment, the mAP was mainly used as the evaluation index of model detection accuracy, and FPS was used to evaluate whether the detection speed of the model met the requirements of underwater detection tasks. The threshold of the IOU was set to 0.5.

### 3.3. Experimental Results

The relevant parameters of the model needed to be set before the experiment. The SGD (Stochastic Gradient Descent) method was applied to optimise the learning parameters during model training; the weight decay was set to 0.0005, and the momentum was set to 0.8. The resolution of the input image was set to 640 × 640; the batch size was set to 56; the learning rate was set to 0.01, and the training threshold of the IOU (Intersection Over Union) was set to 0.2. The correlation coefficients hue (H), saturation (S), and lightness (V) for data augmentation were set to 0.015, 0.7, and 0.4, respectively. During the experiment, it was found that if there are too many training epochs, the model is subjected to overfitting, resulting in a decrease in the detection accuracy of the model. Therefore, in this experiment, we set the training epochs to 90. Under such settings, the mAP value of the model reached the highest value and tended to be stable, and overfitting did not occur at this time. Figure 9 shows the loss curves of the modified YOLOv5 model on the training and validation sets during the training process. As seen in Figure 9, when the model was trained for 90 epochs, the loss curves tended to converge. When model training was completed, the corresponding weight and related files were saved for subsequent target detection tasks and related indicator evaluations.

A comparison was made with three benchmark models to verify the effectiveness of the model proposed in this paper. The used models were YOLOv3, YOLOv4, and YOLOV5s. All models were set with the same parameters, trained in the same environment, and tested on the same underwater dataset. The test results are recorded in Table 1.

It can be seen in Table 1 that the mAP of the modified model proposed in this paper (80.90% mAP) was 2.4% higher than that of the original YOLOv5s (78.50% mAP) model, 15.58% higher than that of the YOLOv4 model (65.32% mAP), and 36.84% higher than that of the YOLOv3 model (44.06% mAP). However, as seen in Table 1, in the category of echinus, there was not much difference in the AP of the four models. Among the underwater datasets used, the echinus dataset has the largest sample size, with approximately 10,000 samples, which is much larger than the sample sizes of the other three datasets. Having a sufficient number of samples can significantly enhance the detection performance of the model. In this study, in order to meet the hardware restriction requirements of micro underwater robots, we utilized the simplest model in the YOLOv5 series. Although the YOLOv5s model has lower detection accuracy than other complex models, it is preferred due to its simplicity. However, when the sample size is substantial, the YOLOv5s model can be customized to improve detection accuracy. Nevertheless, the accuracy achieved with this modification may not surpass that of more complex models. However, the modified model had relatively balanced detection accuracy for each type, and the detection accuracy was not too low due to the small sample size. In contrast, the AP values of the YOLOv5s model and the modified YOLOv5s-CA in the three categories of starfish, scallop, and holothurian were higher than those of the YOLOv3 and YOLOv4 models. Although the AP of the original YOLOv5s model in the holothurian category was 0.2% higher than that of the YOLOv5s-CA model, the overall effect was not significantly affected. This is mainly because the modification of the original YOLOv5s model increased the complexity of the model, which may have reduced the performance of the detection of some specific types. It can be concluded, according to Table 1, that the proposed YOLOv5s-CA showed the best performance in the underwater target detection tasks.

To verify the real-time performance of the modified model, the FPS value of each model was tested, and the test results are recorded in Table 2. As can be seen in Table 2, when tested on the same device, the YOLOv3 model had the fastest detection speed (79 FPS); YOLOv4 had the slowest detection speed (52 FPS); and the detection speed of the modified model proposed in this paper was 71 FPS, indicating that the modified model can meet the real-time requirements of underwater target detection tasks. It was also found that the proposed YOLOv5s-CA improved the underwater detection accuracy while guaranteeing a satisfactory detection speed.

In Figure 10, the detection performance of the modified model proposed in this paper is shown in four different scenarios: (a) an explicit scene at close range, (b) a blurred scene at close range, (c) a clear scene from a long distance, and (d) a blurred scene from a long distance. In this paper, the detection threshold was set to 0.25, meaning that only when the target detection score was greater than or equal to 0.25, the model showed detection accuracy, detection box, and type of target. When the detection threshold was set to 0.25, the model could detect the vast majority of targets in the image. It can be seen in Figure 10 that the modified YOLOv5s model could effectively detect the targets in the four scenarios, showing that the model could achieve satisfactory detection performance in complex underwater environments.

Some comparative experiments were conducted to show that the model proposed in this paper can accurately perform target detection tasks in complex underwater environments. Figure 11 shows the corresponding test results, where (a) is the original underwater image, (b) shows the detection result using the YOLOv3 model, (c) shows the detection result using the YOLOv4 model, (d) shows the detection results using the YOLOv5s model, and (e) shows the detection result using the YOLOv5s-CA model proposed in this paper. In Figure 11, it can be seen that the model proposed in this paper, compared with the three other models, could comprehensively detect blurred targets. The modified model proposed in this paper improved the detection accuracy by guaranteeing real-time performance.

Because the proposed target detection model is mainly used in micro underwater robots, which have strict hardware constraints, the parameters and calculations of the model need to be limited. In this paper, YOLOv5s-CA was compared with YOLOv5m, YOLOv5l, YOLOv5x, and YOLOv7. The comparison results are shown in Table 3. It can be seen in Table 3 that the YOLOv5s-CA model achieved better detection performance while maintaining a low amount of parameters and calculations.

In summary, it can be concluded that embedding CA attention and SE attention modules into the YOLOv5s model can improve the accuracy of underwater target detection while guaranteeing good real-time detection performance in complex underwater environments. The experimental results show that the proposed YOLOv5s-CA achieved the best accuracy among the YOLOv3, YOLOv4, and YOLOv5s models, demonstrating the proposed algorithm.

## 4. Conclusions

Because of the poor performance of underwater robots in the detection of specific underwater targets due to the complex underwater environment, this paper proposes a modified YOLOv5s model, which was applied to actual underwater tasks. By embedding the CA attention module and SE attention module into the YOLOv5s model, the accuracy of the modified model in the detection of blurry underwater objects was improved. The experimental results show that the method proposed in this paper has good advantages in underwater target detection. In the future, more effective attention modules with image enhancement could be an interesting research topic for the accuracy improvement in the detection of blurred underwater targets.

## Figures and Tables

**Figure 1 sensors-23-03367-f001:**
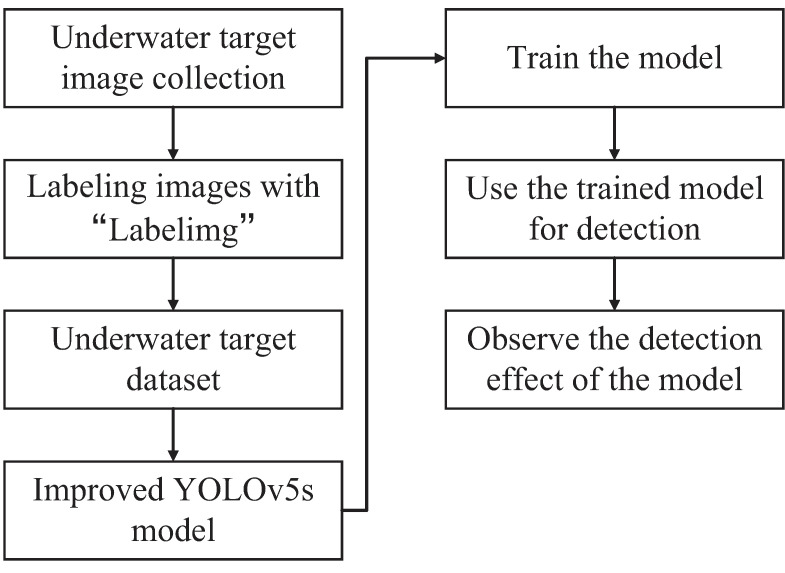
Diagram of process for applying the modified YOLOv5s model to underwater target detection.

**Figure 2 sensors-23-03367-f002:**
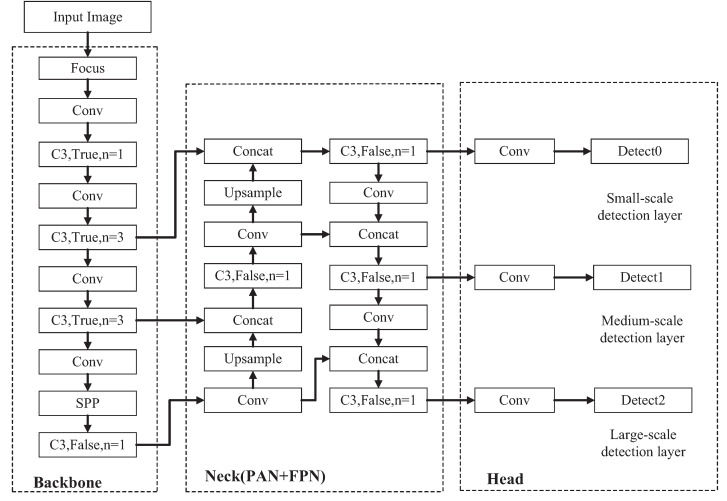
Overall network structure of the YOLOv5s model. The backbone network convolves the input image and converts it into feature maps of different sizes. The neck network fuses high-level and low-level feature maps, and the fused feature maps are input into the detect network to detect objects in the picture.

**Figure 3 sensors-23-03367-f003:**
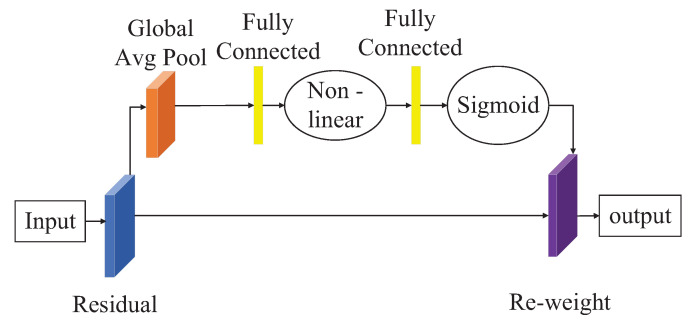
Structure of the SE attention module.

**Figure 4 sensors-23-03367-f004:**
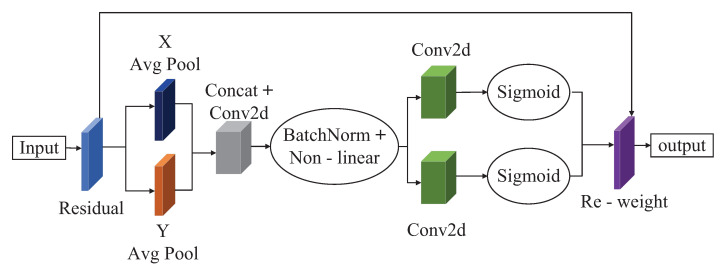
Structure of the CA attention module.

**Figure 5 sensors-23-03367-f005:**
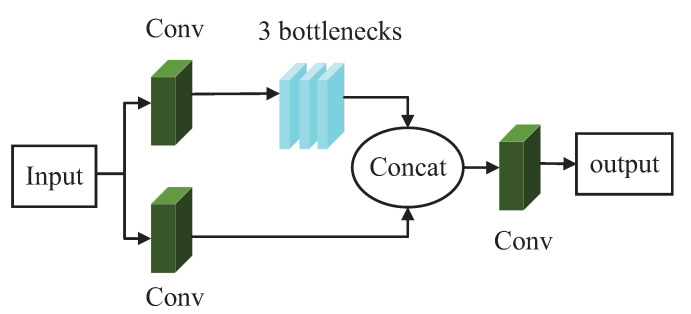
Schematic diagram of the first C3 module here improved.

**Figure 6 sensors-23-03367-f006:**
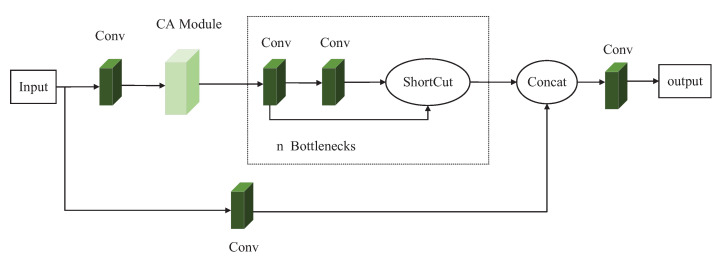
Schematic diagram of the C3-1 module.

**Figure 7 sensors-23-03367-f007:**
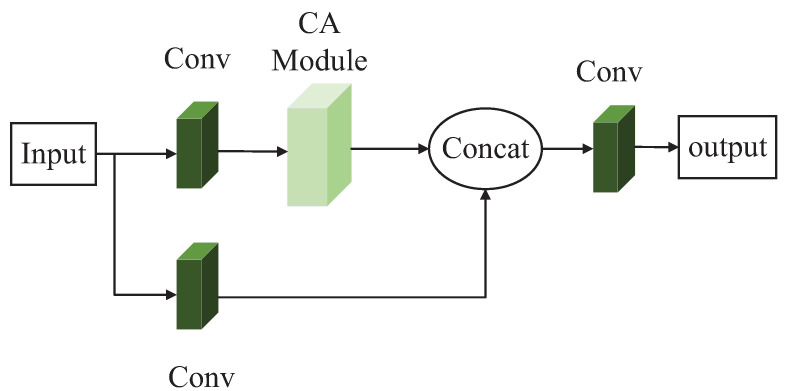
Schematic diagram of the C3-2 module.

**Figure 8 sensors-23-03367-f008:**
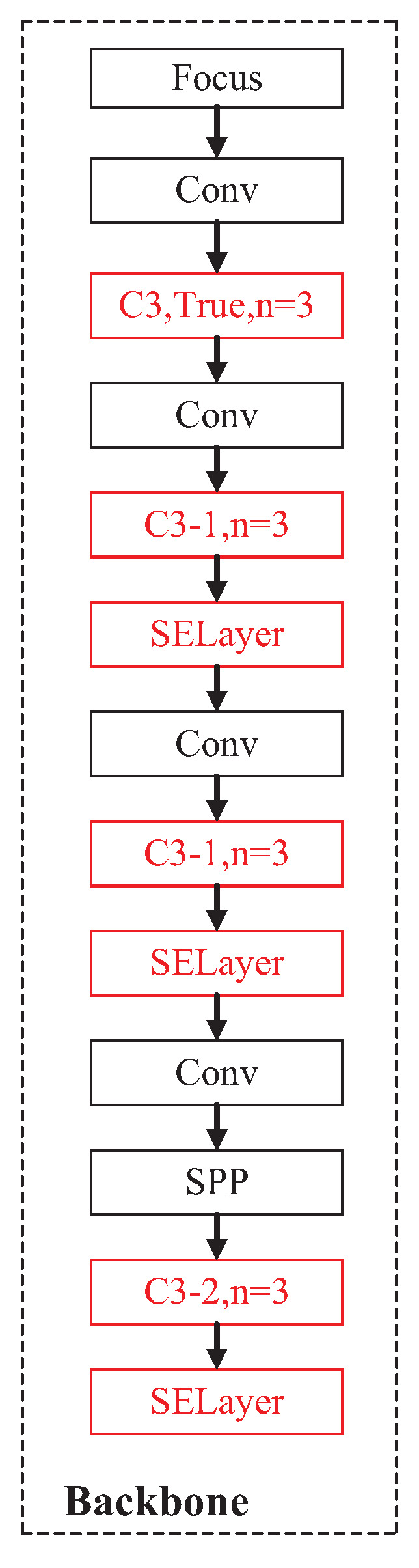
Structural diagram of the modified backbone network.

**Figure 9 sensors-23-03367-f009:**
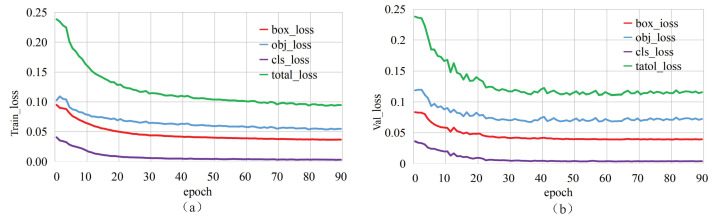
Loss curves of the modified YOLOv5 model during the training process: (**a**) Training set; (**b**) Validation set.

**Figure 10 sensors-23-03367-f010:**
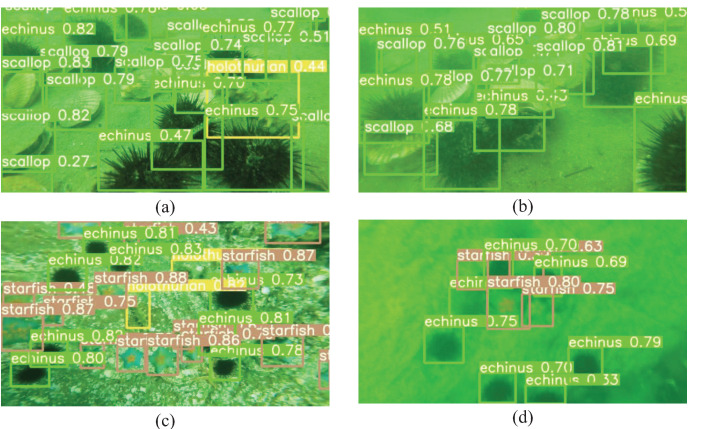
Detection results of different underwater scenes. (**a**) Clear scene at close range, (**b**) blurred scene at close range, (**c**) clear scene from a long distance, and (**d**) blurred scene from a long distance.

**Figure 11 sensors-23-03367-f011:**
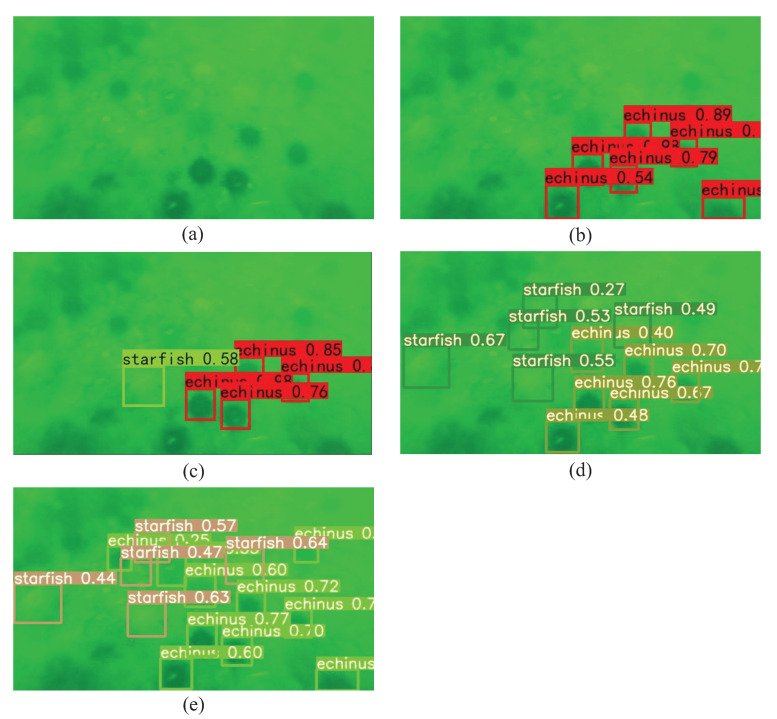
Detection results of different detection models. (**a**) Original underwater image, (**b**) detection result of the trained YOLOv3 model, (**c**) detection result of the trained YOLOv4 model, (**d**) detection result of the original YOLOv5s model, and (**e**) detection result of the YOLOv5s-UW model proposed in this paper.

**Table 1 sensors-23-03367-t001:** Average Precision of each model on the underwater dataset (%).

Module	mAP	Echinus	Starfish	Scallop	Holothurian
YOLOV3	44.06	88.02	61.67	11.48	15.06
YOLOV4	65.32	88.80	79.49	46.38	46.59
YOLOV5s	78.50	84.60	87.40	69.00	73.00
Ours	80.90	86.40	88.30	76.20	72.80

**Table 2 sensors-23-03367-t002:** Detection speed of each model on the underwater dataset.

Module	YOLOV3	YOLOV4	YOLOV5s	Ours
FPS	79	52	76	71

**Table 3 sensors-23-03367-t003:** Results of comparative experiments.

Module	mAP (%)	Parameters (M)	GFLOPS
YOLOv5m	78.9	21.1	50.4
YOLOv5l	77.9	46.6	114.3
YOLOV5x	78.6	87.3	217.4
YOLOv7	80.3	37.2	104.8
YOLOv5s-CA	80.9	7.8	18.6

## Data Availability

The data presented in this study are available on request from the corresponding author.

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
