# Peer review of "YOLOv5s-CA: A Modified YOLOv5s Network with Coordinate Attention for Underwater Target Detection"

_sensors, 2023, doi:10.3390/s23073367_

Round 1

Reviewer 1 Report

This paper proposes a modified YOLOV5s deep learning network for underwater target detection. A coordinate attention module and a squeeze-and-excitation module are added in the network to concentrate on computing power while improving detection accuracy. The proposed algorithm is applied for underwater target detection, and experimental results show the effectiveness of the algorithm. This paper is well written and the developed algorithm makes sense to the aquaculture. Here are some comments to improve this paper: 

1. In this work, SE module is the key method applied to improve detection accuracy. However, the literature review on the concentrate attention modules is not sufficient. Please specify what is the advantages of the used SE module comparing to the other CA modules, e.g., SK.

2. Bottleneck modules are increased from one to three to improve the feature extraction capability. Please explain why the number of bottleneck modules are three but not others.

3. Please explain why add the SE attention module at the output of the C3 module but not elsewhere.

4. Some references appeared in the References list are not cited in the paper.

Author Response

Thanks for the valuable comments, which greatly help to improve our work. Please kindly find our response to the comments in the attached document.

Reviewer 2 Report

The paper proposes an improvement to the YOLOv5s Network for underwater image detection by adding multiple Squeeze-and-Excitation Networks layers and coordination attention mechanisms. While the paper is generally well-written and easy to understand, there are a few areas where it could be improved.

Firstly, the paper contains some explanations that may be too elaborate for those already familiar with the topic. For example, the YOLOv5s model architecture is well-known and there may not be a need to explain it in detail. Instead, highlighting the differences between the proposed model and YOLOv5s in Fig 8 would be sufficient. Similarly, a concise explanation and highlighting of the differences in the C3 module would be more appropriate.

Secondly, in the results section, Table 1 shows slightly lower performance for the proposed model when detecting Echinus and Holothurian. The authors could provide a logical explanation for this inconsistency to help readers understand why this occurred. Additionally, Table 2 shows a lower FPS for the proposed scheme compared to some other models, and the authors could explain why this is the case.

Overall, the paper has potential but could benefit from streamlining and providing logical explanations for performance inconsistencies.

Author Response

(The authors gave the same response as above.)

Reviewer 3 Report

The authors have proposed YOLOv5s model, for the underwater detection of objects.  The CA 365 attention module and SE attention module integration have improved the detection capability, which is the advantage of the proposed work.

The overall structure is good and authors are advised to recheck for language and grammatical errors.

Best wishes to the authors.

Author Response

(The authors gave the same response as above.)

Reviewer 4 Report

This paper proposes a modified YOLOv5s network, called the YOLOv5s-CA network, by embedding a Coordinate Attention (CA) module and a Squeeze-and-Excitation (SE) module for underwater target detection. The manuscript is well-organized. However, I recommend a major revision in this round due to some existing problems weaken the overall quality, the authors are expected to address these issues as follows:

1.    In Eq.(3) uses the letter R, which has the same letter as that of Eq.(2), but obviously the two R have different meanings and are ambiguous.

2.    Figure 9 is too light and looks not clear. It is recommended that the lines be bold and the font color be deepened.

3.    There are many variants of YOLOV5, such as YOLOV5s, YOLOV5m, YOLOV5l, YOLOV5x and also YOLOV7. Why Yolov5s is selected? It is suggested to supplement the results of the comparative experiment.

Author Response

(The authors gave the same response as above.)

Round 2

Reviewer 4 Report

I carefully read through the revised manuscript, and it has revised according to my review opinions. I recommend accepting.